Protocol

# Elements of organisation of integrated maternity care and their associations with outcomes: a scoping review protocol

Jolanda Liebregts ![ORCID],[1] Bahareh Goodarzi,[1] Pim P Valentijn ![ORCID],[2] Soo Downe,[3] Jan Jaap Erwich ![ORCID],[4] George Burchell,[5] Ronald Batenburg,[6] Eline F de Vries,[7,8] Ank de Jonge,[9] Corine J M Verhoeven,[1,10] VOICE study group

For numbered affiliations see end of article.

**Correspondence to**
Ms Jolanda Liebregts;
j.d.liebregts@amsterdamumc.nl

## ABSTRACT

**Introduction** Integrated care is seen as an enabling strategy in organising healthcare to improve quality, finances, personnel and sustainability. Developments in the organisation of maternity care follow this trend. The way care is organised should support the general aims and outcomes of healthcare systems. Organisation itself consists of a variety of smaller 'elements of organisation'. Various elements of organisation are implemented in different organisations and networks. We will examine which elements of integrated maternity care are associated with maternal and neonatal health outcomes, experiences of women and professionals, healthcare spending and care processes.

**Methods and analysis** We will conduct this review using the JBI methodology for scoping reviews and the reporting guideline PRISMA-ScR (Preferred Reporting Items for Systematic Reviews and Meta-Analysis extension for Scoping Reviews). We will undertake a systematic search in the databases PubMed, Scopus, Cochrane and PsycINFO. A machine learning tool, ASReview, will be used to select relevant papers. These papers will be analysed and classified thematically using the framework of the Rainbow Model of Integrated Care (RMIC). The Population Concept Context framework for scoping reviews will be used in which 'Population' is defined as elements of the organisation of integrated maternity care, 'Context' as high-income countries and 'Concepts' as outcomes stated in the objective of this review. We will include papers from 2012 onwards, in Dutch or English language, which describe both 'how the care is organised' (elements) and 'outcomes'.

**Ethics and dissemination** Since this is a scoping review of previously published summary data, ethical approval for this study is not needed. Findings will be published in a peer-reviewed international journal, discussed in a webinar and presented at (inter)national conferences and meetings of professional associations.

The findings of this scoping review will give insight into the nature and effectiveness of elements of integrated care and will generate hypotheses for further research.

## INTRODUCTION

Many countries face challenges in organising healthcare in terms of finances, personnel

## STRENGTHS AND LIMITATIONS OF THIS STUDY

⇒ We will use ASReview, an open source, Artificial Intelligence (AI)-aided software, which will allow us to efficiently select the relevant papers from a large amount of literature found in the systematic search.
⇒ The Rainbow Model of Integrated Care will be used as an explanatory framework to see what evidence is available on which level of integrated care and subsequently which information is missing.
⇒ The members of the research team and expert group working on this scoping review represent different views in maternity care; this ensures a broad scope and reduction of bias.
⇒ We will focus on the organisation of integrated maternity care in high-income countries and exclude literature from low-income and middle-income countries; this is a limitation for drawing general, global conclusions.

and sustainability. Integrating different echelons of care is seen as an enabling strategy in this. Developments in maternity care follow this trend. The topic of integrated maternity care is complex, and the literature shows different definitions of integrated care. We adapted the definition from Allana *et al*[1] to integrated maternity care: 'when the network(s) of multiple professionals and organisations across the maternity care and social care systems provide accessible, comprehensive and coordinated services to women who want to get pregnant, are pregnant and/or are up to 6 weeks after birth'. The way integrated maternity care is organised should support the general aims and outcomes of healthcare systems. Generally accepted goals of healthcare systems are improving population health, enhancing client experiences and reducing costs per person (triple aim).[2,3] Improving healthcare professionals' experiences has subsequently

been added as well (quadruple aim).[4] Healthcare goals can also be formulated in terms of sustainability, that is, seeking the best outcomes for clients and the population against optimal use of resources (time, carbon, use of space; value-based healthcare).[5] The organisation of integrated maternity care is a means to achieve these goals. However, due to different perspectives, cultures and interests of stakeholders, organising care is complex. The organisation itself consists of a variety of smaller 'organisational elements', including different models of risk selection, group-based antenatal care, continuity of care and different payment systems. Different elements of the organisation of care are implemented in different organisations and networks, while little is known about their effects' effects.[6] Furthermore, integrated care concerns different levels of the organisation: organisation at client level (guidelines and protocols), between professionals (who does what and when?), within healthcare organisations and healthcare organisations on a systemic level (payment system, legislation, etc). We will use the Rainbow Model of Integrated Care (RMIC) to obtain a systemic understanding of the design, delivery, management and evaluation of the organisation of integrated care.[7] We will examine 'What makes integrated maternity care work' not only by identifying elements of organisation of integrated care and their associations with outcomes but also by mapping the circumstances and conditions under which integrated maternity care can contribute to the aims and outcomes of care. The aim of this scoping review is therefore to gain a better understanding of the amount and type of evidence on (a) elements of the organisation of integrated maternity care and (b) their association with maternal and neonatal health,[2 5 8] experiences of women and professionals,[8] healthcare spending[2 5] and care processes.[5] A preliminary search of MEDLINE, the Cochrane Database of Systematic Reviews and JBI Evidence Synthesis was conducted, and no completed ongoing systematic reviews or scoping reviews on the topic were identified.

### Review question

Review question: Which elements of the organisation of integrated maternity care are associated with maternal and neonatal health outcomes, experiences of women and professionals, healthcare spending and care processes?

### Eligibility criteria

In our scoping review, we will use the population-context-concept framework, recommended in the JBI manual for Scoping Reviews.[9]

### Population

The 'Population' in this review consists of the various elements of the organisation of integrated maternity care. We will explore the type and level of evidence found for these elements, their associations with outcomes as well as their distribution on the different levels of the RMIC framework.

### Context

The 'Context' is limited to high-income countries because this study is designed to inform practice in high-income countries. In addition, the organisation of care in low-income and middle-income countries is very different. This implies that it would be meaningful to conduct a separate review in these settings.

### Concept

The 'Concepts' in this review are maternal and neonatal outcomes,[5] experiences of clients[10] and professionals,[8] care delivery processes[11 12] and healthcare spending.[2 8]

### Types of studies

This scoping review will consider both experimental and quasiexperimental study designs including systematic reviews, meta-analysis, randomised controlled trials, non-randomised controlled trials, before and after studies and interrupted time-series studies. In addition, analytical observational studies including prospective and retrospective cohort studies, case-control studies and cross-sectional studies will be considered for inclusion. We will also consider descriptive observational study designs for inclusion such as case series and cross-sectional studies. Qualitative studies based on, but not limited to, phenomenology, grounded theory, ethnography, action research and feminist research will be considered as well. In addition, policy documents will be considered.

## METHODS

The proposed scoping review will be conducted using the JBI methodology for scoping reviews[13] and the reporting guideline PRISMA- ScR (Preferred Reporting Items for Systematic Reviews and Meta-Analysis extension for Scoping Reviews); see online supplemental file 1 for the PRISMA-ScR checklist.[14] We will analyse the literature using in-depth reading and analysis, in which we will use Valentijn's RMIC as an organising framework for the elements of the organisation of integrated maternity care.[7] The RMIC is chosen over other frameworks because it offers an open-ended, exploratory, broad view of integrated care that can be applied to various, diverse elements of the organisation of maternity care. This will ensure that consideration will be given to different types of integration (clinical, professional, organisational, system, functional and normative) at multiple levels (micro, meso and macro) and varying degrees (linkage, coordination, full integration).

In our study, the research team will cooperate with a team of experts, members of the VOICE study group, who are involved in different levels of maternity care who will advise on all aspects of the study. To ensure a diversity of perspectives within the expert group, the following representatives will be included: professionals, representatives from national professional associations (ie, client representatives, the Dutch Organisation of Midwives (KNOV), the Dutch Society of Obstetricians (NVOG), the Dutch

Association of Nurses and Carers (V&VN), Knowledge Centre Maternity Care (KCKZ), representatives from health insurance companies, policymakers (ie, Dutch College of Perinatal Care (CPZ)) and professional educational institutes and experts on organisation science, maternity care, epidemiology and economic analysis. The team of experts will be further referred to as the 'expert group'.

## Search strategy

An initial limited search of PubMed was undertaken to identify articles on the topic. The text of titles and abstracts of relevant articles and the keywords used to describe the articles will be used to develop a full search strategy for PubMed, Scopus, PsycINFO and Cochrane Library with the help of an information specialist (see online supplemental file 2). We will consult the expert group to develop the final comprehensive literature search.

The search strategy, including all identified keywords and index terms, will be adapted for each included database and/or information source by the information specialist. The reference list of all included sources of evidence will be screened for additional studies.

Studies published in Dutch and English language will be included as reviewers are fluent in both Dutch and English. We will include papers from 2012 onwards, since this is the year the Member States of the WHO European Region endorsed the European health policy *Health 2020*, acknowledging health system strengthening as one of four priority action areas and calls for people-centred health systems.[6] This generated attention to the organisation of integrated care.

The databases to be searched include PubMed, Scopus, PsycINFO and Cochrane Library. Sources of unpublished studies and grey literature to be searched include Google Scholar, key websites of interest (ie, WHO, government agencies), hand searches of reference lists of papers and prior and additional knowledge of the expert group.

We will focus on (a) elements of organisation of integrated maternity care; (b) outcomes of maternal and neonatal health, women's and professionals' experiences, healthcare processes and healthcare costs; and (c) levels of integrated care (RMIC). Potentially relevant references will further be obtained from the retrieved publications and by consulting the expert group (snowball method).

Identified citations will be collected and uploaded into Endnote 20,[15] and duplicates will be removed. The inclusion and exclusion criteria will be applied based on a priori decisions made by the research team. Reviewers JL and BG will do the selection of relevant papers in two phases.

For the first selection of relevant papers, based on the title and abstract, we will use an open-source machine learning-aided pipeline applying active learning: ASReview,[16] V.1.1. This machine learning tool needs to be 'trained' to effectively rank the papers based on possible relevance. ASReview will be trained based on 'prior knowledge': papers the research team and expert group considers to be relevant to the topic at hand. In the active learning cycle, the model incrementally improves its predictions on the remaining unlabelled records, by which relevant records are identified as early in the process as possible.

To determine when to stop screening, we elaborated on the stopping criterion for this screening, based on both knowledge within the research team and heuristic arguments as time and resources available. We will stop screening when a maximum of 400 papers are labelled relevant or 50 papers sequentially are labelled as irrelevant.

In the first stage, titles and abstracts will be screened by two independent reviewers (JL, BG) for assessment against the inclusion criteria for the review (table 1). To ensure inter-reviewer agreement in both title and abstract screening as well as full-text screening, JL and BG discuss uncertainties and specify and adjust the inclusion and exclusion criteria. In phase 2, the full-text screening will be done by JL and BG. Any disagreements that arise between the reviewers at each stage of the selection process will be resolved through discussion or with a third reviewer (CV).

After we selected the relevant papers, an export file (.csv) will be extracted out of ASReview and will be presented to the research team and expert group. In these files, both the labelled papers (papers labelled as relevant or irrelevant) and unlabelled papers will be visible. They are put in order of relevance based on the machine learning tool.

When this first selection of relevant papers is finished, full texts and their citation details will be retrieved and imported into ASReview. The full text of selected papers will be assessed in detail against the inclusion and exclusion criteria by two independent reviewers (JL and BG) (table 1). Reasons for exclusion will be recorded and reported in the scoping review. The results of the search and the study inclusion process will be reported in full and presented in line with the PRISMA-ScR flow diagram (figure 1).[14]

Relevant papers will contain information on elements of the organisation AND one or more of the outcomes of interest.

## Data extraction

Data will be extracted from papers included in the scoping review by two independent reviewers (JL and BG) using a data extraction tool developed by the research team. The data extracted include specific details about (a) the elements of the organisation of integrated maternity care in high-income countries; (b) specific details on associations of the elements of the organisation on maternal and neonatal health, experiences of clients and healthcare professionals, care processes and healthcare spending; and (c) the taxonomy provided by the RMIC will be used to extract data on different levels of integrated care according to the RMIC framework.[17] Also, data will

**Table 1** Inclusion and exclusion criteria for selecting relevant papers.

| Inclusion | Exclusion |
|---|---|
| Elements of organisation of integrated maternity care | Elements NOT related to the organisation of care, that is, content of care |
| Network(s) of multiple professionals and organisations across the maternity care and social care systems | |
| Information on accessible, comprehensive and coordinated services in maternity care | |
| Midwives, obstetricians, obstetric nurses/nurse midwives, maternity care assistants and their clients/patients (people who want to get pregnant, are pregnant and/or are up to 6 weeks after birth) | Articles in which either midwives or obstetricians play no role at all |
| Maternal and neonatal health; pregnancy outcomes, morbidity, mortality and complications | |
| Patient experiences, healthcare professional experiences, well-being, satisfaction | |
| Healthcare spending; costs, expenditure -> money and assets -> that is, more or fewer healthcare professionals/hours, etc | |
| Care processes; referral, intervention, transfer, medicalisation | |
| Peer-reviewed papers, articles in professional journals, policy documents, RCTs, cohort and case-control studies, case studies, qualitative research, editorials, comments | Abstracts of congresses and congress presentations, research protocols |
| Papers published between 2012 and 2022 | Papers published before 2012 |
| Maternity care in high-income countries | Low-income and middle-income countries Healthcare is other than maternity care. Non-regular context -> refugees, war situation, pandemic |
| Language: English and Dutch | Other languages than English and Dutch |

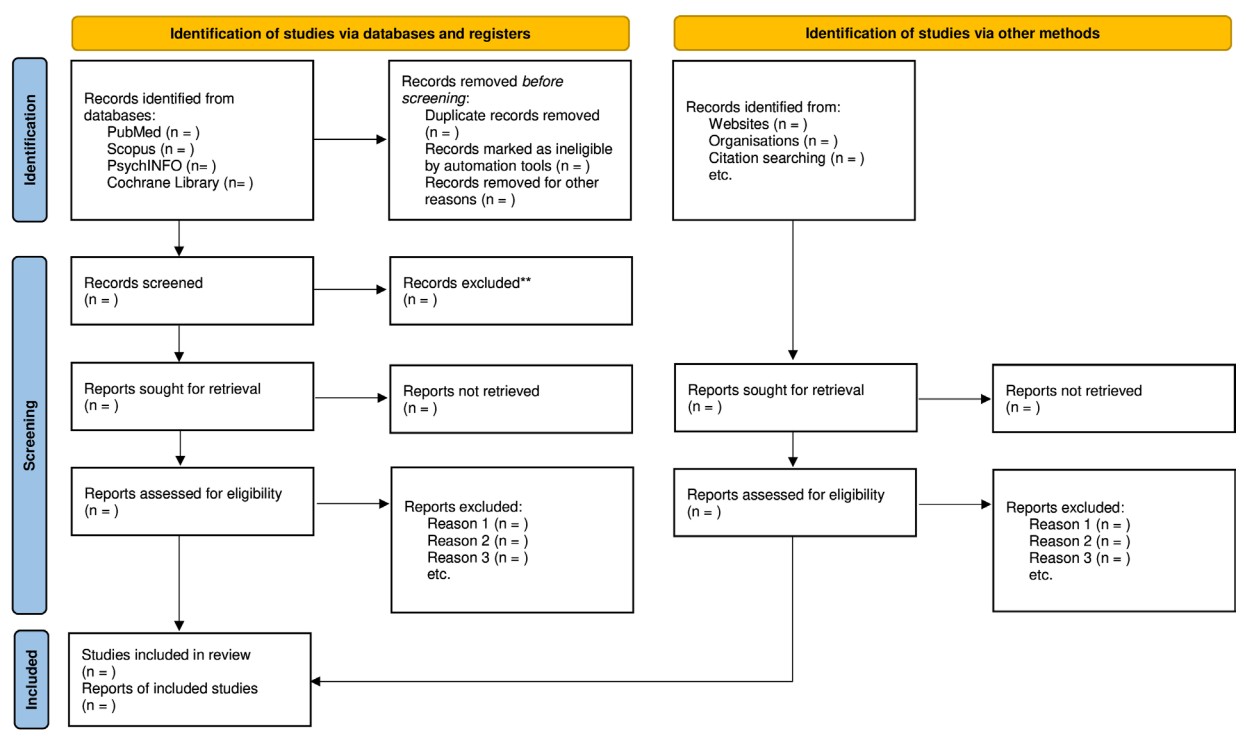

*Consider, if feasible to do so, reporting the number of records identified from each database or register searched (rather than the total number across all databases/registers).
**If automation tools were used, indicate how many records were excluded by a human and how many were excluded by automation tools.

**Figure 1** Flow diagram of the study.[19]

be categorised based on the methods used in the included papers to analyse the level of evidence per element of organisation.

For the data extraction, we will use a method based on the framework analysis[18] using the framework of RMIC to identify the different elements of organisation of integrated maternity care, their associations with outcomes and the level of integrated care they belong to.

A draft extraction form is provided in online supplemental table 1). The draft data extraction tool will be modified and revised as necessary during the process of extracting data from each included evidence source. Modifications will be detailed in the article with the results of the scoping review

We will follow the same procedure as previously described in the literature selection. The research team and expert group determined the criteria for the data extraction table (online supplemental table 1). Two reviewers (JL and BG) will extract data independently out of three times 20 randomly selected included papers and discuss until they reach a complete agreement. Any disagreements that arise between the reviewers JL and BG will be resolved through discussion or with a third reviewer (CV). Full data extraction will only be started after reviewers have obtained sufficient agreement. If appropriate, authors of papers will be contacted to request missing or additional data, where required.

## Data analysis and presentation

Data will be presented initially in a regular table based on the data extraction form (table 1). Depending on the nature of emerging associations, we plan to use a range of graphical solutions to illustrate the findings.

## Patient and public involvement

Patients and/or the public were not involved in the design, or conduct, or reporting, or dissemination plans of this research.

**Author affiliations**
[1]Midwifery Science, Amsterdam UMC Locatie VUMC, Amsterdam, Netherlands
[2]Department of Health Services Research, Maastricht University, Maastricht, Netherlands
[3]Research in Childbirth and Health, University of Central Lancashire, Lancashire, UK
[4]Obstetrics and Gynaecology, University of Groningen, Groningen, Netherlands
[5]Medical Library, Vrije Universiteit Amsterdam, Amsterdam, Netherlands
[6]Netherlands Institute for Health Services Research, Utrecht, Netherlands
[7]Department of Quality of Care and Health Economics, Centre for Nutrition, Prevention and Health Services, Bilthoven, Netherlands
[8]Leiden University Campus The Hague, National Institute of Public Health and Primary Care, The Hague, Netherlands
[9]Midwifery Science, AVAG, APH Research Institute, Amsterdam UMC Locatie VUMC, Amsterdam, Netherlands
[10]Midwifery, School of Health Sciences, University of Nottingham, Nottingham, UK

**Acknowledgements** This review is part of a PhD trajectory of Liebregts, JD.

**Collaborators** In addition to the authors of this scoping review, the VOICE study group comprises Irene de Graaf, Puck van Heemstra, Hester Rippen, Jeroen Struijs, Susanne Zuidhof, Inge Boesveld, Anouk Kaiser, Mirjam Fransen, Durk Berks, Pauline Haga, Katarzyna Burzynska, Joland Vermolen, Lucienne Bakker, Mariëtte Hoogsteder, Lillian Peters, Conny Vreugdehil, Suzanne Thompson, Aimée Sparendam-Bruijnincx, Betty de Vries, Elle Struijf, Lillianne van der Velde, Bert Horlings, Marianne Nieuwenhuijze, Marc Roosenboom and Simone Plaizier.

**Contributors** JL, BG, PPV, JJE, GB, RB, EFdV, SD, ADJ and CJMV contributed to the development of this study protocol. JL, ADJ and CJMV conceptualised the research question, designed the study and prepared the first draft of the study protocol. BG, SD, JJE, ADJ and CJMV provided methodological expertise. PV provided expertise on the Rainbow Model of Integrated Care. RB and EFdV provide expertise in funding, research and organisation of integrated maternity care. GB is an information specialist at Amsterdam UMC and contributes his expertise in the systematic search and reporting according to PRISMA-ScR. All authors have contributed to the study design and revising the protocol. All authors have approved the final manuscript. All authors agree to be accountable for all aspects of the work in ensuring that questions related to the accuracy or integrity of any part of the work are appropriately investigated and resolved. The VOICE study group was part of the conceptualisation of the VOICE study grant proposal.

**Funding** This scoping review is part of the Dutch VOICE study – Variations in Organization of Integrated CarE, which is funded by ZonMW, the Dutch association for health research and development, program 'Zwangerschap en geboorte II', grant number 05430052110002.

**Competing interests** None declared.

**Patient and public involvement** Patients and/or the public were not involved in the design, or conduct, or reporting, or dissemination plans of this research.

**Patient consent for publication** Not applicable.

**Provenance and peer review** Not commissioned; externally peer reviewed.

**ORCID iDs**
Jolanda Liebregts http://orcid.org/0009-0002-1729-145X
Pim P Valentijn http://orcid.org/0000-0002-9788-1275
Jan Jaap Erwich http://orcid.org/0000-0003-1362-4501

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
