## [Reviewer comments · BMJ Open]

ARTICLE DETAILS

TITLE (PROVISIONAL)	Elements of Organization of Integrated Maternity Care and their Associations with Outcomes: A Scoping Review Protocol
AUTHORS	Liebregts, Jolanda; Goodarzi, Bahareh; Valentijn, Pim; Downe, Soo; Erwich, Jan Jaap; Burchell, George; Batenburg, Ronald; de Vries, Eline; de Jonge, Ank; Verhoeven, Corine; Study Group, VOICE

VERSION 1 – REVIEW

REVIEWER	Siaw-Frimpong, Moses Komfo Anokye Teaching Hospital, Anaesthesia and Intensive Care
REVIEW RETURNED	14-Sep-2023

GENERAL COMMENTS	REVIEW REPORT	
	REVIEW CHECKLIST	RESPONSE
	1 Is the research question or study objective clearly defined?	Yes
	2 Is the abstract accurate, balanced and complete?	Yes
	3 Is the study design appropriate to answer the research question?	Yes
	4 Are the methods described sufficiently to allow the study to be repeated?	Yes
	5 Are research ethics (e.g. participant consent, ethics approval) addressed appropriately?	Yes
	6 Are the outcomes clearly defined?	Yes
	7 If statistics are used, are they appropriate and described fully?	Not applicable
	8 Are the references up-to-date and appropriate?	Yes
	9 Do the results address the research question or objective?	Not applicable. The study is yet to be done.

	10	Are they presented clearly?	Not applicable
	11	Are the discussion and conclusions justified by the results	Not applicable
	12	Are the study limitations discussed adequately?	Generally, yes. My only issue is captured in my comment below.
	13	Is the supplementary reporting complete (e.g. trial registration; funding details; CONSORT, STROBE or PRISMA checklist)?	Yes
	14	To the best of your knowledge is the paper free from concerns over publication ethics (e.g. plagiarism, redundant publication, undeclared conflicts of interest)?	Yes
	15	Is the standard of written English acceptable for publication?	Yes
COMMENT			
The reasons adduced for the exclusion of LMICS and low resource countries are fair. However, if the intention of the study is to give a global picture, then the exclusion of LMICs becomes a limitation and must be stated.			

REVIEWER	Chen, An Aalto-yliopisto Tuotantotalouden laitos, Department of Industrial Engineering and Management
REVIEW RETURNED	18-Sep-2023

GENERAL COMMENTS	Thank you for inviting me to review this protocol. In general this is a high-quality protocol, with an interesting topic. Here are some suggestions.  1. please check from the author guidelines of the BMJ open whether citations could be present in abstract or not. 2. I understand PRISMA provides standards and recommendations of reporting review work. PRISMA is not a guidance of conducting review. Please check and elaborate it precisely. 3. Page 5, Line 40, realityReality 4. check space between lines, e.g., page 7, line 3-17 5. Page 7, Types of sources - Types of studies 6. I would like to know how would you assess the quality of included studies and how you evaluate the confidence of your findings at the end.
--

REVIEWER	Crequit, Simon
-----------------	----------------

	Centre Hospitalier Intercommunal de Montreuil, Gynecology & Obstetrics
REVIEW RETURNED	30-Sep-2023

GENERAL COMMENTS	I first thank the authors to have the pleasure to review the relevant protocol “Elements of Organization of Integrated Maternity Care and their Associations with Outcomes: A Scoping Review Protocol”. Indeed, in developed countries maternity care organization is becoming a major health issue given the association between professional’s well-being, organization of care and pregnancy outcomes. Furthermore, this topic has never been addressed before and there is a clear need for a systematic review on the topic. The research question is clearly defined, and the protocol is clear. My only question is how the authors plan to address heterogeneity on the population of mothers regarding maternal and neonatal outcomes? Indeed these outcomes are highly correlated to the risk level of the pregnancy (tertiary care units versus primary care) and to maternal history. Nonetheless, this research question is crucial, and this protocol should be published and conducted. Here are little English mistakes that should be addressed: Abstract, Line 17, The organization of care is a means Line 24 circumstances and conditions that are enabling Introduction, L23 Organization of care is a means
---

VERSION 1 – AUTHOR RESPONSE

Reviewer: 1	
The reasons adduced for the exclusion of LMICS and low-resource countries are fair. However, if the intention of the study is to give a global picture, then the exclusion of LMICs becomes a limitation and must be stated	We added this limitation to the “strengths and limitations” section on p4 and 5:  We will focus on organization of integrated maternity care in high income countries and exclude literature from low and middle income countries; this a limitation for drawing general, global conclusions
Reviewer: 2	
please check from the author guidelines of the BMJ open whether citations could be present in abstract or not.	We reviewed the author guidelines and consequently removed the references from the abstract.
I understand PRISMA provides standards and recommendations of reporting review work. PRISMA is not a guidance of conducting review. Please check and elaborate it precisely	We have updated the description of and reference to the reporting guidelines PRISMA-ScR (P3 L12 and P9 L13)
Page 5, Line 40, realityReality	When rewriting the Introduction-section, this sentence was dropped.
check space between lines, e.g., page 7, line 3-17	We adjusted the line spacing from 1 to 1.5 (P8, L3-5 and L18-29))

Page 7, Types of sources - Types of studies	We have changed Types of sources to Types of studies (P8 L30)
I would like to know how would you assess the quality of included studies and how you evaluate the confidence of your findings at the end.	We will describe the types of study designs and their findings on associations between elements of integrated care and various outcomes. However, we will not systematically assess the quality of included studies as this is beyond the scope of a scoping review.
Reviewer: 3	
My only question is how the authors plan to address heterogeneity on the population of mothers regarding maternal and neonatal outcomes? Indeed these outcomes are highly correlated to the risk level of the pregnancy (tertiary care units versus primary care) and to maternal history.	Our research will focus primarily on the association between the elements of organization and outcomes/ experiences. If different associations will be found for different groups of women, this will be reported in the Results section. However, addressing heterogeneity of the population is not one of our primary aims.
Here are little English mistakes that should be addressed: Abstract, Line 17, The organization of care is a means Line 24 circumstances and conditions that are enabling Introduction, L23 Organization of care is a means	The new versions of the submissions have been checked by a colleague for the use of the English language. You will find the adjustments in the use of the English language marked in the revised version.